# Comparison of Soil Bacterial Communities under Canopies of *Pinus tabulaeformis* and *Populus euramericana* in a Reclaimed Waste Dump

**DOI:** 10.3390/plants12040974

**Published:** 2023-02-20

**Authors:** Huping Hou, Haiya Liu, Jinting Xiong, Chen Wang, Shaoliang Zhang, Zhongyi Ding

**Affiliations:** 1Engineering Research Center of Ministry of Education for Mine Ecological Restoration, China University of Mining and Technology, Xuzhou 221116, China; 2School of Public Policy and Management, China University of Mining and Technology, Xuzhou 221116, China; 3School of Environment Science & Spatial Informatics, China University of Mining and Technology, Xuzhou 221116, China; 4Observation and Research Station of Ecological Restoration for Chongqing Typical Mining Areas, Ministry of Natural Resources, Chongqing Institute of Geology and Mineral Resources, Chongqing 401120, China

**Keywords:** ecological restoration, bacterial community, high-throughput sequencing, molecular ecological network

## Abstract

To compare the effects of different remediation tree species on soil bacterial communities and provide a theoretical basis for the selection of ecosystem function promotion strategies after vegetation restoration, the characteristic changes in soil bacterial communities after *Pinus tabulaeformis* and *Populus euramericana* reclamation were explored using high-throughput sequencing and molecular ecological network methods. The results showed that: (1) With the increase in reclamation years, the reclaimed soil properties were close to the control group, and the soil properties of *Pinus tabulaeformis* were closer to the control group than those of *P. euramericana*. (2) The dominant bacteria under the canopies of *P. tabulaeformis* and *P. euramericana* was the same. *Proteobacteria*, *Actinobacteria*, *Acidobacteria*, *Chloroflexi*, *Gemmatimonadetes*, *Planctomycetes*, *Bacteroidetes*, and *Cyanobacteria* were the dominant bacteria in the restored soil, accounting for more than 95% of the total abundance. The average values of the Shannon diversity index, Simpson diversity index, Chao 1 richness estimator, and abundance-based coverage estimator of the bacterial community in the *P. euramericana* reclaimed soil were higher than those in the *P. tabulaeformis* reclaimed soil. The influence of reclamation years on the bacterial community of samples is greater than that of species types. (3) The results of ecological network construction showed that the total number of nodes, total number of connections, and average connectivity of the soil bacterial network under *P. euramericana* reclamation were greater than those under *P. tabulaeformis* reclamation. The bacterial molecular ecological network under *P. euramericana* was more abundant. (4) Among the dominant bacteria, the relative abundance of *Actinobacteria* was negatively correlated with soil pH, soil total nitrogen content, and the activities of urease, invertase, and alkaline phosphatase, while the relative abundance of *Proteobacteria* and *Bacteroidetes* was positively correlated with these environmental factors. The relationship between the soil bacterial community of *P. tabulaeformis* and *P. euramericana* and the environmental factors is not completely the same, and even the interaction between some environmental factors and bacteria is opposite.

## 1. Introduction

In 2019, the Ministry of Natural Resources of the People’s Republic of China successively issued the Opinions on Establishing an Incentive Mechanism to Accelerate the Promotion of Ecological Restoration of Mines and the Opinions of the Ministry of Natural Resources on Exploring and Utilizing Marketization to Promote Ecological Restoration of Mines, to accelerate the promotion of ecological restoration of mines and solve outstanding problems, such as many historical arrears, many practical contradictions, and insufficient investment in ecological restoration of mines. It can be seen that promoting land reclamation and ecological restoration is an important way to repair damaged mine ecological functions and realize the sustainable development of the mining ecosystem [1]. From 2001 to 2019, China’s cumulative mine ecological restoration area reached more than 1 million hectares, and in 2020, the newly increased mine restoration area was approximately 48,000 hectares [2]. However, the effect of the ecosystem after ecological restoration and the resilience and adaptability of the ecosystem will be the focus of future research [3]. Soil microorganisms play an important role in maintaining ecosystem stability and vegetation succession restoration [4]. Soil bacterial communities play an important role in regulating soil nutrient cycling and plant species coexistence. The composition and structural changes of soil bacterial communities are often used to reflect the changes in soil environmental quality [5]. Therefore, the analysis of the characteristics of the soil microbial community after reclamation can provide a basis for decision making to improve the effectiveness of the soil ecosystem after reclamation.

Currently, research on the soil bacterial community after reclamation mainly focuses on the following points: first, research on the change in soil microbial characteristics after restoration. Sun et al. studied the response of soil microorganisms to vegetation restoration in coal mining subsidence areas and microbial changes [6]. Li Yuanyuan et al. studied the impact of surface subsidence on soil microbial diversity [7]. Luo et al. suggested that while mining subsidence was inhibited, vegetation rehabilitation promoted the soil physicochemical properties [8]. Dimitriu et al. suggested that the structure of soil bacteria after vegetation rehabilitation was dependent on pH and other abiotic characteristics [9]. Soil bacteria could serve as sensitive indicators of land degradation and ecological restoration [10,11]. Second, the characteristics of soil microorganisms in different remediation years: William et al. and Wanglong et al. studied the changes in soil biological communities in different recovery and evolution stages [12,13]; Hou et al. studied the change rule of soil microbial community structure in reclamation areas with reclamation years [14]. After 20 years of land reclamation in the coal mining area, the number of soil bacteria was still lower than that in an undisturbed area, but shrub coverage played a key role in ecological restoration [15]. After 5–14 years of vegetation rehabilitation, significant interactions were observed between plants and bacteria [16]. The third is about the impact of different reclamation methods on soil microbial community restoration. The diversity and composition of bacterial and fungal communities play an irreplaceable role in decomposition and nutrient cycling [17,18]. Previous studies showed that significant differences existed in the relative abundance of bacteria and fungi after restoration [19,20,21]. Zhang Lin et al. studied the impact of different plant combination planting methods on the soil bacterial community in the reclamation area [22]. *Medicago sativa L*. (*Alfalfa*) and *Bromus inermis Leyss.* (*Smooth brome*) are widely used as a community-building species for ecological restoration in northern China [19,23,24]. *Alfalfa*, a high-quality perennial legume, could improve the soil’s texture and nutrients with a low degree of degradation [25,26]. A short-term vegetation restoration significantly increased the complexity and stability of fungi ecological networks, but the opposite was the case with the bacteria. which confirmed that ecological restoration by sowing was favorable to the amelioration of soil fungi complexity and stability in the short term [27].

However, there are few studies on the changes in soil microorganisms in different vegetation configuration modes and different reclamation years. In the study of soil microorganisms, traditional research methods, such as various diversity indices, can only excavate the composition and abundance information of microbial communities in the surface layer; the amount of microbial information data obtained using high-throughput sequencing is rich and huge. For this reason, in this study, we selected soil with *Pinus tabulaeformis* and *Populus euramericana* as the reclamation tree species in the different waste dumps of the Heidaigou open-pit coal mine as the study area and used high-throughput technology to obtain soil microbial information. Furthermore, we used the molecular ecological network to reveal and compare the changes and characteristics of the bacterial communities in the soil reclaimed using *P. tabulaeformis* and *P. euramericana* and compared the effects of different vegetation reclamation models on soil bacterial communities, providing a theoretical basis for the selection of vegetation reclamation.

## 2. Materials and Methods

### 2.1. Overview of the Study Area

The study area is located in the Heidaigou open-pit coal mine in the Zhungeer Coalfield, east of Zhungeer County, Ordos City, Nei Monggol Autonomous Region, with coordinates 111°12′53″–111°20′02″ E and 39°43′11″–39°47′41″ N (Figure 1). The mining area is located east of Ordos. The main geomorphic type is low mountains and hills, and the overall terrain is high in the west and low in the east. The branch ditches of the Heidai Valley and Dajiaoshao Valley are the main valleys in the area. The mining area has a semi-arid continental climate in the middle temperate zone. The soil is mainly castanozems, the zonal vegetation is typical grassland, and most of the artificial forests are sparse forests made up of vegetation types, such as *P. tabulaeformis*, *P. euramericana*, and *Platycladus orientalis* (L.) *Francoptmxjjkmsc*. The farmland is mainly dry and dispersed. The surface is directly damaged due to open-pit mining. The method of landform reconstruction and vegetation reconstruction is adopted for restoration. The soil cover is sourced from topsoil stripped from open-cut mining, with a thickness of approximately 1.0 m. The soil is mainly loess, characterized by a relatively infertile, paucity of humus, and has not undergone any processes of maturation. The vegetation species to be restored are mainly *P. tabulaeformis* and *P. euramericana*.

This sampling object was used to select the reclaimed land under different restoration years and different tree species. The north waste dump (15 years of reclamation), the east and west waste dump (12 years of reclamation), the inner waste dump (9 years of reclamation), and the Yinwan waste dump (6 years of reclamation) after the restoration of the Heidaigou open-pit coal mine were selected as the sampling sites, and the sites not affected by coal mining subsidence were selected as the control sites. The locations of the sampling sites are shown in Figure 1.

### 2.2. Sample Collection and Analysis

From 17–21 July 2019, sample sites rehabilitated using *P. tabulaeformis* and *P. euramericana* were set up in five waste dump sites; the control site, which was not affected by mining, was set up with *P. tabulaeformis* and *P. euramericana* (Table 1). Five samples were collected from each sample area, and a total of 60 samples were collected. Five hundred grams of 0–10 cm mixed soil samples was collected using the five-point sampling method. The ring knife was used for measuring the soil bulk density. The soil samples were collected with a sterile shovel. After being mixed evenly, part of the soil samples was put into an aluminum box for measuring the soil moisture, part was put into a special sterile sealed bag, and part was put into a sterile test tube. The soil samples in the test tube were immediately put into the vehicle-mounted refrigerator for storage at −20 °C and then mailed to Shanghai Personalbio Biotechnology Co., Ltd. (Shanghai, China) for high-throughput sequencing.

A pH meter (Alipay Biotechnology Co., Ltd., Hangzhou, China) was used to measure the soil pH; a Topsizer laser particle size analyzer (OMCC Instrument Co., Ltd., Zhuhai, China) was used to measure soil clay, silt, and sand contents; and an Automatic Kjeldahl nitrogen meter (Qingdao Jingcheng Instrument Co., Ltd., Qingdao, China) was used to measure soil total nitrogen. Furthermore, an ultraviolet absorption spectrophotometer (Qingdao Jingcheng Instrument Co., Ltd., Qingdao, China) was used to measure the soil total phosphorus content, soil organic matter content, soil alkali hydrolyzable nitrogen, soil urease, soil alkaline phosphatase, and soil sucrase activity. Soil total potassium was measured using a flame spectrophotometer (Shanghai Changxi Instrument & Meters Co., Ltd., Shanghai, China).

### 2.3. DNA Extraction and High-Throughput Sequencing

The 16S rRNA genes of soil bacteria were sequenced within three days of sampling. The DNA of soil samples was extracted using the E.Z.N.A^®^ Soil DNA Kit (Omega Bio-tek, Norcross, GA, USA) according to the manufacturer’s protocols. The V4–V5 area of the 16S rRNA of soil bacteria was amplified via polymerase chain reaction (PCR). The applied degenerate primers were 515F 5′-barcode-GTGCCAGCMGCCGCGG-3′ and 907R 5′-CCGTCAATTCMTTT RAGTTT-3′.

The PCR amplification products were detected using electrophoresis, the target fragments were cut and recycled, and the fluorescence quantification was performed based on the electrophoresis detection results. The TruSeq Nano DNA LT Library Prep Kit of Illumina Company was used to prepare the sequencing library and to perform purification, quality inspection, and quantification. The qualified library was diluted by gradient and mixed according to the required sequencing amount in the corresponding proportion, and finally, the MiSeq Agent Kit V3 (600 cycles) was used as the reagent for computer sequencing. The optimal sequencing length of the target fragment was 200–450 bp.

### 2.4. Molecular Ecological Network Analysis

Based on the reclaimed vegetation, the samples were divided into two groups: *P. tabulaeformis* and *P. euramericana*, and the OTU with the highest abundance of 75% was selected for each group to construct and analyze the molecular ecological network. The construction process of the ecological network is found in MENA (http://ieg4.rccc.ou.edu/mena, 15 August 2019). The specific analysis process is as follows: High-throughput sequencing data were obtained to determine the number of different OTUs for each sample; the relevant abundance of OTU numbers was standardized. The Spearman correlation coefficient was used to calculate the pairwise similarity of each OTU and to generate the similarity matrix. Based on the random matrix theory (RMT), the derived threshold was determined, and the adjacency matrix was constructed.

After the construction of the molecular ecological network, the topological structure and modular analysis of the network were performed. Topological analysis was used to verify whether the results of network construction meet the scale-free, small-world, and modular characteristics of molecular ecological networks, and to test the reliability of the construction results [28]. Modularization analysis was used to judge the role of nodes in the network by calculating and comparing the connectivity between modules and connectivity within modules of nodes, dividing the functions of network nodes [29], and finally, visualizing the molecular ecological network using Cytoscape software.

### 2.5. Data Analysis and Processing

Python’s matplotlib package was used to draw a bar graph of bacterial abundance at the phylum level and a heat map of bacterial abundance at the class level in the soil samples. The Pearson correlation between soil physical and chemical properties and enzyme activity was analyzed using SPSS 25.0. Common indices for calculating alpha diversity include the Simpson diversity index, Shannon diversity index, Chao 1 richness index, and abundance-based coverage estimator (ACE) richness index. Principal component analysis (PCA) and nonmetric multidimensional scaling (NMDS) were used to reflect the beta diversity of the bacterial communities.

## 3. Results

### 3.1. Physical and Chemical Properties and Enzyme Activity Characteristics of Soil under Different Reclamation Tree Species and Different Years

The soil physical and chemical properties of different reclamation tree species are shown in Figure 2. There are certain differences in soil physical and chemical properties and enzyme activities under *P. tabulaeformis* and *P. euramericana*. There is no significant difference in the physical properties of the soil under the two types of plants. The average soil water content under *P. tabulaeformis* is 1.12% higher than that under *P. Euramericana*, while the average values of bulk density and sand content under *P. tabulaeformis* are 0.05 g/cm^2^ and 2.63% lower than that under *P. euramericana*. The average pH value and average cation exchange capacity (CEC) of the soil under *P. tabulaeformis* are higher than those of *P. euramericana* by 0.01 and 1.25 coml/kg, respectively, while the average total nitrogen, total phosphorus, total potassium, and soil organic matter contents are relatively low by 0.01%, 0.06 g/kg, 0.33 g/kg, and 1.89 g/kg, respectively. The activities of soil urease, sucrase, and alkaline phosphatase under *P. tabulaeformis* are 0.18 mg/g, 3.72 mg/g, and 0.22 mg/g lower than those under *P. euramericana*, respectively. After 15 years of reclamation, the soil organic matter content and urease activity were significantly different from the control group, and bulk density, soil water content, pH, cation exchange capacity, total nitrogen, total phosphorus, total potassium, and invertase activity were basically close to the control group. In short, with the increase in reclamation years, the reclaimed soil properties were close to those of the control group, and the soil properties of *P. tabulaeformis* were closer to those of the control group than those of *P. euramericana*.

With the increase in reclamation years, the soil bulk density gradually decreased from 2.45 g/cm^2^ to 2.17 g/cm^2^, and the proportion of soil sand particles gradually increased from 64.32% to 75.29%, while the soil water content had no obvious change. The soil pH gradually decreased from 8.29 to 8.2 with the increase in reclamation years. The cation exchange capacity, soil organic matter, total nitrogen, and total potassium content increased significantly with the increase in reclamation years, from 11.5 coml/kg, 7.09 g/kg, 0.03%, and 12 g/kg to 14.45 coml/kg, 11.73 g/kg, 0.05%, and 12.35 g/kg, respectively. The total phosphorus of the soil did not change significantly with the increase in reclamation years. The activities of urease, sucrase, and alkaline phosphatase increased significantly with the increase in reclamation years, from 0.13 mg/g, 8.44 mg/g, and 0.52 mg/g to 0.47 mg/g, 45.26 mg/g, and 0.81 mg/g, respectively.

### 3.2. Characteristics of Bacterial Community Structure and Diversity in the Reclaimed Soil of P. tabulaeformis and P. euramericana

#### 3.2.1. Characteristics of the Bacterial Community Structure in the Reclaimed Soil of *P. tabulaeformis* and *P. euramericana*

The bacterial community composition of each sample at the phylum level was statistically analyzed (Figure 3). At the phylum level of the samples from the waste dump of the Heidaigou open-pit coal mine and the control area, the bacterial community with high abundance mainly includes *Proteobacteria*, *Actinobacteria*, *Acidobacteria*, *Chloroflexi*, *Gemmatimonadetes*, *Planctomycetes*, *Bacteroidetes*, and *Cyanobacteria*, accounting for more than 95% of the total abundance.

The relative abundance of *Proteobacteria* was the largest, ranging from 27.34% to 39.34%. In 12 a and 15 a of reclamation, the relative abundance of *Proteobacteria* in *P. euramericana* was higher than that of *P. tabulaeformis*. However, the relative abundance of *P. tabulaeformis* was lower than that of *P. euramericana* in 6 a and 9 a of reclamation. *Proteobacteria* participate in the soil nitrogen cycle process and help to restore the improvement of soil quality. The relative abundance of *Actinobacteria* was 16.55–36.82%, which was increasing with the increase in reclamation years. *Actinobacteria* was a saprophytic aerobic bacterium that preferred neutral or slightly alkaline, promoting the growth of rhizosphere bacteria, symbiotic bacteria, and endophytic bacteria. The relative abundance of *Acidobacterium* was 7.32–17.12%. In the samples of *P. tabulaeformis*, the abundance was increasing with the increase in reclamation years. This bacterium plays an important role in the degradation of plant residues. The abundance of *Chloroflexi* in *P. tabulaeformis* samples decreased with the increase in reclamation years. The nutrition mode of *Chloroflexi* was extremely diverse. From the perspective of vegetation type, at the phylum level, the relative abundance of *Actinobacteria* in the reclaimed soil of *P. tabulaeformis* is 0.04% higher than that in the reclaimed soil of *P. euramericana*. Furthermore, the relative abundance of *Acidobacteria* and *Cyanobacteria* in the reclaimed soil of *P. tabulaeformis* is 0.02% lower than that in the reclaimed soil of *P. euramericana*.

From the perspective of the changing trend of bacterial community structure, *Proteobacteria*, *Actinobacteria*, *Acidobacteria*, *Gemmatimonadetes*, and *Cyanobacteria* have a high response to the change in reclamation years. The relative abundance of *Proteobacteria*, *Gemmatimonadetes*, and *Cyanobacteria* decreased by 2.43%, 3.73%, and 7.61% on average each year, while the relative abundance of *Actinobacteria* and *Acidobacteria* increased by 8.31% and 3.08% on average each year.

#### 3.2.2. Diversity of the Bacterial Community in the Reclaimed Soil of *P. tabulaeformis* and *P. euramericana*

The alpha diversity index can measure the community diversity of samples. At the same time, the Simpson diversity index, Shannon diversity index, Chao 1 richness estimator index, and ACE were selected for comparative analysis of soil bacterial community diversity in different vegetation reclamation models and different reclamation years. As shown in Figure 4, the Shannon index of the two tree species has a significant difference (*p* < 0.1), while the differences in other indices are not significant (*p* > 0.1), but the average values of the four indices of the bacterial community in the reclaimed soil of *P. tabulaeformis* are higher than those in that of *P. tabulaeformis*.

#### 3.2.3. Beta Diversity of the Bacterial Community in the Reclaimed Soil of *P. tabulaeformis* and *P. euramericana*

Beta diversity analysis can measure the similarity between different community composition samples. The PCA method based on the Euclidean distance was used to analyze the OTU of each sample, and the results are shown in Figure 5a. The PCA1 axis can explain 35.55% of the difference in the results of the sample, and the PCA2 axis can explain 15.73% of the difference in the results of the sample. There are obvious differences between the sample points (C, W, E, and N) with longer reclamation years and the sample points (Y and I) with shorter reclamation years (Figure 5a). The distance between the samples with the same reclamation years is closer, indicating that they are more similar. The sample points of the same reclamation vegetation are scattered, indicating that the impact of the reclamation vegetation type on the soil bacterial community is less than the impact of the reclamation years. The difference in bacterial community composition with *P. tabulaeformis* as the reclamation vegetation is relatively larger than that with *P. euramericana* as the reclamation vegetation.

Based on the phylogenetic tree, we comprehensively considered the changes in species and species abundance, calculated the weighted unifrac distance between samples, and performed an NMDS analysis. The results are shown in Figure 5b. The sample points of *P. tabulaeformis* and *P. euramericana* are scattered, and the distance between samples with the same reclamation years is closer. This may be because the influence of reclamation years on the bacterial community of samples is greater than that of vegetation types, and this result is consistent with the result of the PCA method.

### 3.3. Analysis of Bacterial Molecular Ecological Network in the Reclaimed Soil of P. tabulaeformis and P. euramericana

#### 3.3.1. Topological Structural Analysis of the Molecular Ecological Network

Topological analysis was used to verify whether the results of the network construction meet the scale-free, small-world, and modular characteristics of the molecular ecological network and to test the reliability of the network construction. The samples were divided into the *P. tabulaeformis* group and the *P. euramericana* group, and the number of OTUs in each group was 725 and 818, respectively. Based on the RMT theory, the two groups of data were used to derive an adjacency matrix and to generate a network at the threshold level of 0.960. Under this threshold, the chi-square value of the *P. tabulaeformis* group is 60.779, and that of the *P. euramericana* poplar group is 69.672, which belongs to the appropriate threshold level. The analysis of the network topology (Table 2) shows that the number of ecological network nodes in *P. tabulaeformis* and *P. euramericana* is 642 and 746, respectively. The number of ecological network connections in *P. tabulaeformis* and *P. euramericana* is 1518 and 2003, respectively. The R2 values of both groups are greater than 0.7, which conforms to the power law distribution and reflects the scale-free characteristics of the molecular ecological network. The average connectivity of *P. tabulaeformis* and *P. euramericana* is 4.729 and 5.370, respectively. The average clustering coefficients of *P. tabulaeformis* and *P. euramericana* are 0.541 and 0.579, respectively. The results reflect the small-world and modular characteristics of the molecular ecological network. The average connectivity, average clustering coefficient, and average path distance of the *P. euramericana* group are greater than those of the *P. tabulaeformis* group, indicating that the connection of the soil bacterial molecular ecological network of the *P. euramericana* group is more complex, and the nodes in the network are more closely connected with the adjacent nodes.

#### 3.3.2. Modularization Analysis of the Molecular Ecological Network

The inter- and intra-module connectivity of nodes were calculated and compared to determine the role of nodes in the network. The modular analysis results are shown in Figure 6. The soil bacterial ecological network under *P. tabulaeformis* has six module hubs, of which OTU 19,113 and OTU 22,783 belong to *Chloroflexi*, OTU 12,216 and OTU 30,681 belong to *Planctomycetes*, OTU 8668 belongs to *Proteobacteria*, and OTU 979 belongs to *Actinomycetes*. The soil bacterial ecological network under *P. euramericana* has six module hubs, of which OTU 485, OTU 19,889, and OTU 2,580 belong to *Planctomycetes*, OTU 14,346 and OTU 6293 belong to *Actinobacteria*, and OTU 15,619 belong to *Gemmatimonadetes*. The soil bacterial ecological network under *P. tabulaeformis* has four connection nodes, among which OTU 24,802, OTU 25,594, and OTU 185 belong to *Actinobacteria*, and OUT 79,908 belongs to *Proteobacteria*. The soil bacterial ecological network under *P. euramericana* has one connection node, OTU 5057, which belongs to *Actinobacteria*.

#### 3.3.3. Results of the Molecular Ecological Network Construction

The visualization results directly show the relationship between nodes and modules. After modularization analysis, 61 modules are generated in the *P. tabulaeformis* group, with a module index of 0.862, and 65 modules are generated in the *P. euramericana,* with a module index of 0.841. The module indices of both groups are high, indicating that the two groups of network systems have high stability against external changes. Based on the results of the module analysis, Cytoscape was used to visualize the molecular ecological network (Figure 7). The nodes of the same module are distributed on the same circle, the color of the point represents the phylum types to which it belongs, the connecting line represents the interaction between nodes, the blue connecting line between nodes represents the positive interaction between them, and the red connecting line represents the negative interaction between them.

Comparing the molecular ecological network of soil bacteria under *P. tabulaeformis* and *P. euramericana*, the bacterial molecular ecological network under *P. tabulaeformis* is less than the bacterial molecular ecological network under *P. euramericana* by six modules, indicating that the bacterial molecular ecological network under *P. euramericana* is more abundant. The number of connections between and within the bacterial molecular ecological network modules under *P. euramericana* is greater, and the proportion of red connection lines is higher than that of *P. tabulaeformis*, indicating that the negative interaction between bacterial communities is stronger under *P. euramericana*. The proportion of green connection lines under *P. tabulaeformis* is higher than that of *P. euramericana*, indicating that the positive interaction between bacterial communities is stronger. Therefore, the relationship between bacterial communities under *P. euramericana* is more complex, and the relationship between and within modules is closer.

## 4. Discussion

### 4.1. Differences in Response of Soil Bacterial Communities to Different Vegetation Types

Land reclamation can improve the structure of the soil bacterial community, increase the number and diversity of soil microorganisms, and enhance the abundance of functional genes [30]. However, different vegetation types have different impacts on soil microorganisms in the reclamation area [31], and different types of reclamation vegetation are suitable for different climatic conditions. Li found that the shrub soil in the mining area of the Loess Plateau has better biochemical characteristics and higher microbial diversity than the soil of trees and grasslands, which is more suitable for reclamation [32]. Wang J found that the number of bacteria and fungi in the rhizosphere of trees in the western mining area is higher than that of shrubs and herbs, and the ecological restoration effect is also better [33]. Therefore, the influence of vegetation on soil characteristics is different under different climatic conditions. This study found that there are some differences in soil microbial community diversity and community composition under *P. tabulaeformis* and *P. euramericana*. The results of this study are consistent with those of Helong [34]. *P. euramericana* is conducive to the development of soil bacterial diversity. Therefore, the reclamation vegetation selection of the Heidaigou open-pit waste dump is more conducive to the improvement of soil quality to increase the planting proportion of *P. euramericana*.

### 4.2. Differences in Soil Bacterial Community Response to Different Reclamation Years

From the perspective of reclamation years, the longer the reclamation year, the higher the diversity of soil bacterial communities. The change in the OTU number of each sample also conforms to this law. According to the PCA of each sample at the phylum level, the distance between samples with the same reclamation period is closer, and the similarity between soil samples with the same reclamation period is stronger. The diversity of the soil bacterial community also increases with an increase in reclamation time. For example, Ezeokoli et al. surmised that the structure of soil bacterial communities can reflect the potential differences between different soil ecosystems and that the bacterial diversity and function of reclaimed soil will recover over time [35]. Buta et al. surmised that vegetation restoration can significantly improve soil quality and have a significant impact on the activity of soil microorganisms [36]. Morrien et al. supposed that in the process of reclamation, the relative abundance of *Actinobacteria* and *Acidobacteria* would increase with an increase in reclamation years [37], and the diversity of soil bacterial communities with longer reclamation years was higher [38]. Therefore, the reclamation period is an important factor affecting the restoration of the soil bacterial community in the reclamation area.

### 4.3. Driving Mechanisms of Bacterial Community Structure Change in the Reclaimed Soil

RDA analysis was conducted on environmental factors and the main bacteria in the waste dump samples of the Heidaigou open-pit coal mine (Figure 8). The results show that the relative abundance of Actinobacteria is positively affected by soil alkaline nitrogen content, total nitrogen content, alkaline phosphatase activity, invertase activity, and urease activity. The relative abundance of *Proteobacteria*, *Bacteroidetes*, and *Armatimonadetes* is also strongly negatively affected by these environmental factors. The total phosphorus content of the soil has a significant positive correlation with the relative abundance of *Chloroflexi* and a strong negative correlation with the relative abundance of *Planctomycetes*. CEC is positively correlated with the relative abundance of *Planctomycetes* and negatively correlated with the relative abundance of *Chloroflexi*. The total potassium content of the soil is positively correlated with the relative abundance of *Acidobacteria* and negatively correlated with the relative abundance of *Gemmatimonadetes* and *Cyanobacteria*. The proportions of clay and silt in the soil are negatively correlated with the relative abundance of *Deinococcus-Thermus*.

To explore the interaction between the soil bacterial community and environ-mental factors under *P. tabulaeformis* and *P. euramericana*, the top 200 OTUs of soil bac-terial abundance under *P. tabulaeformis* and *P. euramericana* were selected, and the en-vironmental factors were used as nodes to build an interaction network between the soil bacterial species composition and environmental factors. The results are shown in Figure 9. Proteobacteria and Actinobacteria still occupy the dominant position (Figure 9). In the molecular ecological network diagram (Figure 9a) of soil bacterial species com-position and environmental factors under *P. tabulaeformis*, there are better connections between environmental factors, such as soil water content, CEC, soil total nitrogen, soil organic matter, and soil urease activity, and other nodes. Among them, soil organic matter, CEC, and soil urease activity mainly have positive interactions with bacteria, while soil water content and soil total nitrogen mainly have negative interactions with bacteria. In the molecular ecological network diagram (Figure 9b) of soil bacterial species composition and environmental factors under *P. euramericana*, there are many connections between environmental factors, such as soil total nitrogen, soil organic matter, soil invertase activity, and soil alkaline phosphatase activity, and other nodes. Soil total nitrogen, soil organic matter, soil invertase activity, and soil alkaline phosphatase activity mainly have positive interactions with bacteria. It can be concluded that the relationship between the soil bacterial community of *P. tabulaeformis* and *P. euramericana* and the environmental factors is not completely the same, and even the interaction between some environmental factors and bacteria is opposite, which also shows that the growth environment required by *P. tabulaeformis* and *P. euramericana* is different, which needs further exploration.

## 5. Conclusions

(1) According to the analysis of the structure and diversity of the soil bacterial community, *Proteobacteria*, *Actinobacteria*, and *Acidobacteria* are the main bacteria in the soil. The relative abundance of *Actinobacteria* is greater in the soil samples under *P. tabulaeformis* than in the soil samples under *P. euramericana*, and the relative abundance of *Acidobacteria* is less in the soil samples under *P. tabulaeformis* than in the soil samples under *P. euramericana*. The soil bacterial diversity is relatively higher under *P. euramericana* than under *P. tabulaeformis*, and the difference in bacterial community composition among samples is smaller, which is more conducive to the restoration of soil bacterial diversity. The activities of soil enzymes are higher under *P. euramericana* than under *P. tabulaeformis*.

(2) The total number of nodes, total number of connections, and average connectivity of the soil bacterial network under *P. euramericana* are greater than those under *P. tabulaeformis*. Based on the connectivity between nodes, it can be concluded that the relationship between bacterial communities under *P. euramericana* is more complex, and the relationship between different modules is also closer. Therefore, *P. euramericana* is more conducive to the restoration of soil bacterial diversity than *P. tabulaeformis*.

(3) Among the major bacteria, the relative abundance of *Actinobacteria* is negatively correlated with soil pH, soil total nitrogen content, urease activities, invertase activities, and alkaline phosphatase activities, while the relative abundance of *Proteobacteria*, *Bacteroidetes*, and *Armatimonadetes* is positively correlated with these environmental factors. Soil total nitrogen content, soil organic matter content, and the activities of urease, invertase, and alkaline phosphatase have a greater impact on the bacterial community.

(4) The influence of *P. euramericana* and *P. tabulaeformis* on soil bacterial community structure is related to the mechanisms of plant growth, which we will continue to study in our follow-up work.

Different restoration tree species have different effects on the soil bacterial community and the ecological function of restoration of reclaimed land. Therefore, through the experimental study of the change rule of soil microorganisms, we can timely understand the reasons for soil quality changes and select the appropriate repair tree species, which is more conducive to accelerating the selection of vegetation configuration on the reclaimed land and the speed of ecosystem restoration. However, the differences between the tree species that they derive from have not been studied in depth, and the mechanism will continue to be studied in the future.

## Figures and Tables

**Figure 1 plants-12-00974-f001:**
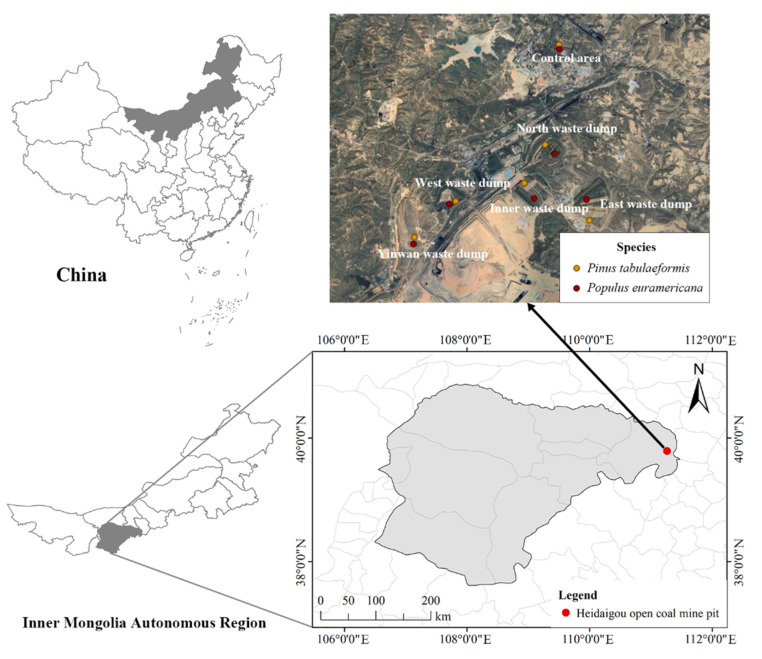
Location of the study area.

**Figure 2 plants-12-00974-f002:**
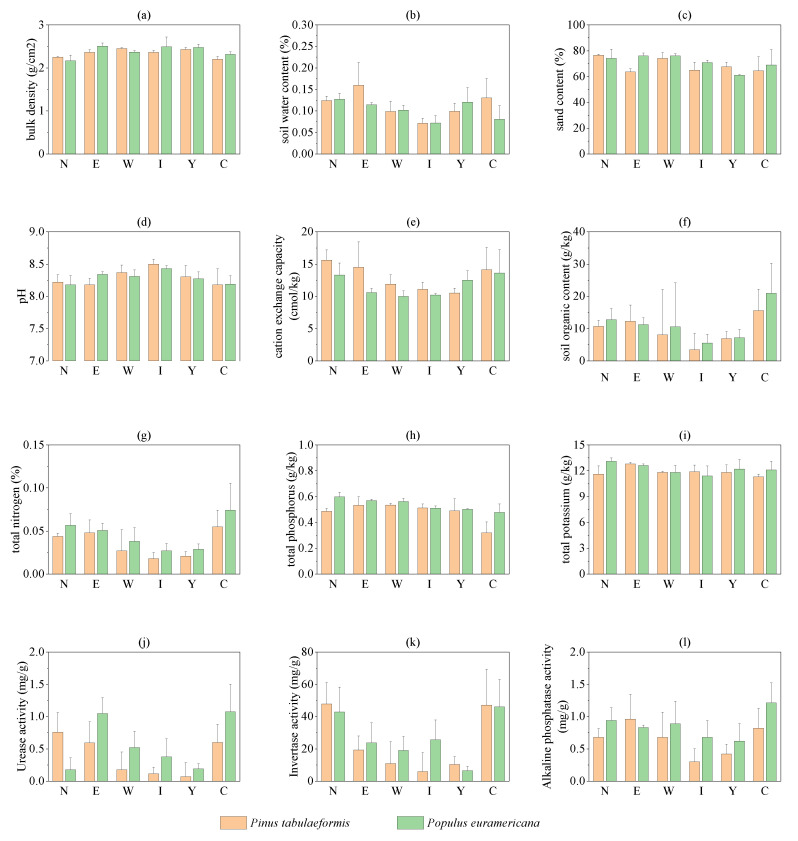
Soil physicochemical properties and enzyme activities. Note: Y is the Yinwang waste dump, I is the inner waste dump, W is the west waste dump, E is the east waste dump, N is the north waste dump, and C is the control site. (**a**) is bulk denisity, (**b**) is soil water content, (**c**) is sand content, (**d**) is PH, (**e**) is cation exchange capacity, (**f**) is soil organic content, (**g**) is total nitrogen, (**h**) is total phosphorus, (**i**) is total potassium, (**j**) is urease activity, (**k**) is invertase activity, (**l**) is alkaline phosphatase activity.

**Figure 3 plants-12-00974-f003:**
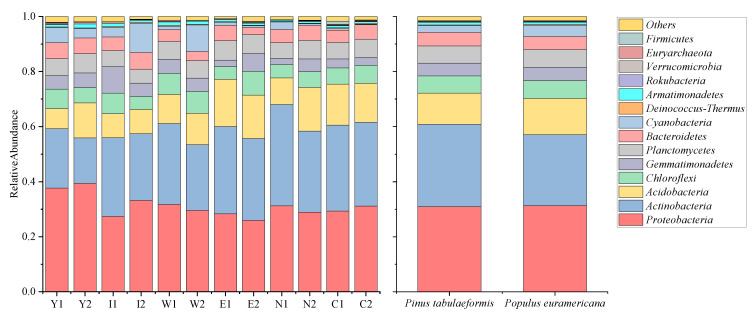
Composition and relative abundance of bacterial community at the phylum level of each sample in the Heidaigou open-pit coal mine dump and control area.

**Figure 4 plants-12-00974-f004:**
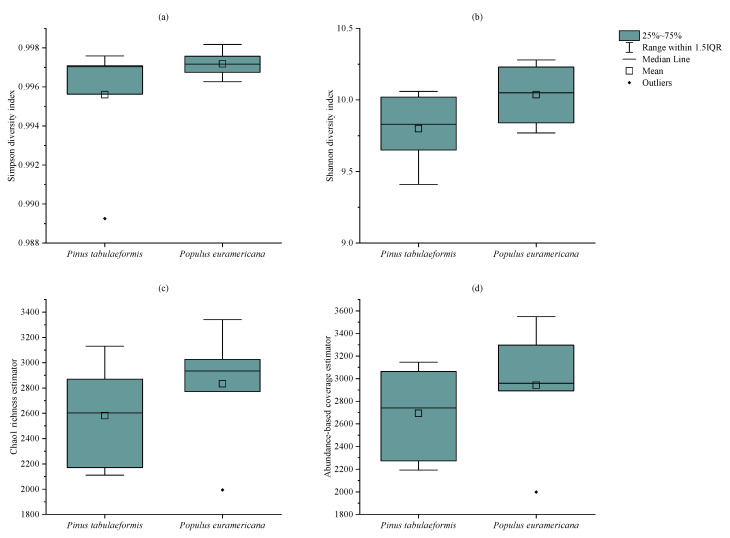
Alpha diversity index of each sample under the reclamation of *P. tabulaeformis* and *P. euramericana.* (**a**) is Simpason diversity index, (**b**) is Shannon diversity index, (**c**) is Chao 1 richness estimator, (**d**) is Abundance based coverage estimator.

**Figure 5 plants-12-00974-f005:**
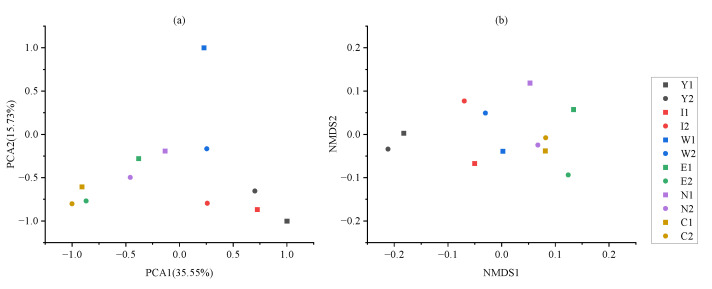
PCA and NMDS analysis of each sample in the Heidaigou open-pit coal mine dump and control area. (**a**) is PCA, (**b**) is NMDS.

**Figure 6 plants-12-00974-f006:**
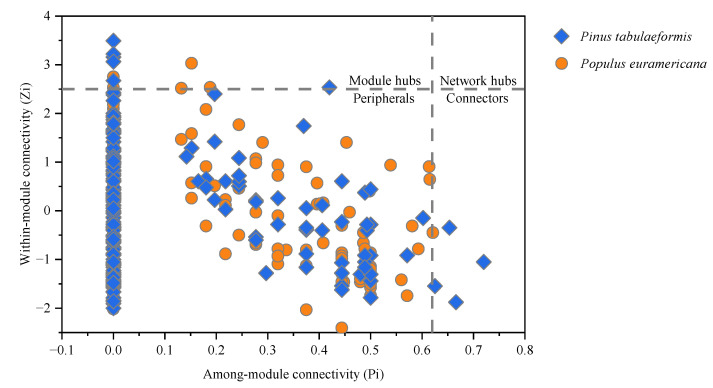
Z-P plot of soil bacterial molecular ecological networks under the reclamation of *P. tabulaeformis* and *P. euramericana*.

**Figure 7 plants-12-00974-f007:**
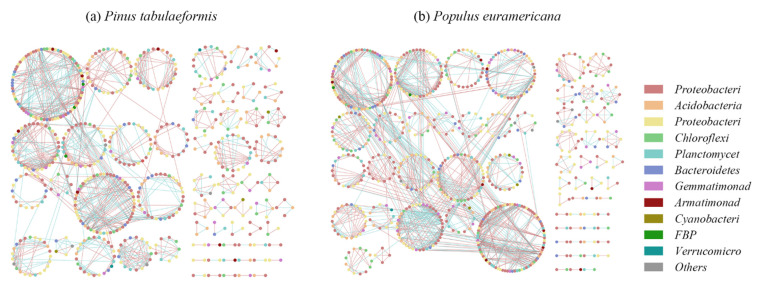
Soil bacterial molecular ecological networks under Pinus tabulaeformis and Populus euramericana.

**Figure 8 plants-12-00974-f008:**
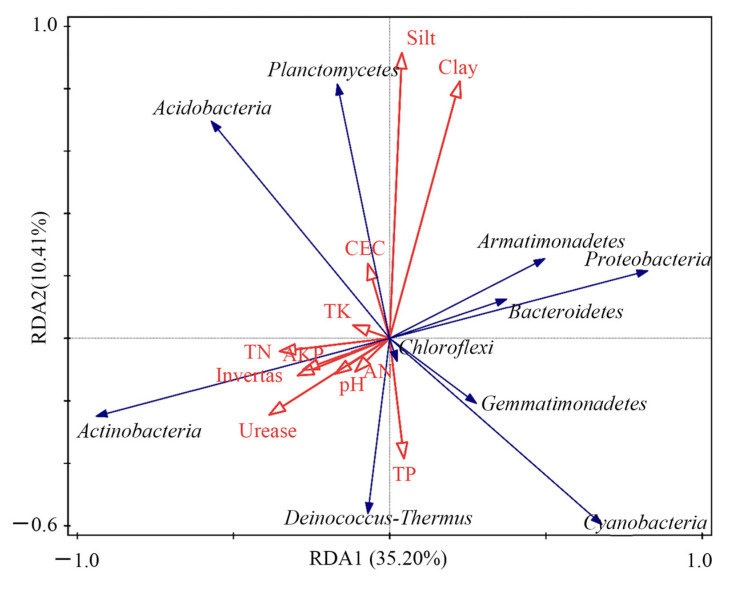
RDA analysis of environmental factors and major bacteria in the Heidaigou open-pit coal mine dump and control area. Note: AKP, soil alkaline phosphatase activity; CEC, soil cation exchange capacity; pH, soil pH; TK, soil total potassium; TN, soil total nitrogen; TP, soil total phosphorus; and WC, soil water content.

**Figure 9 plants-12-00974-f009:**
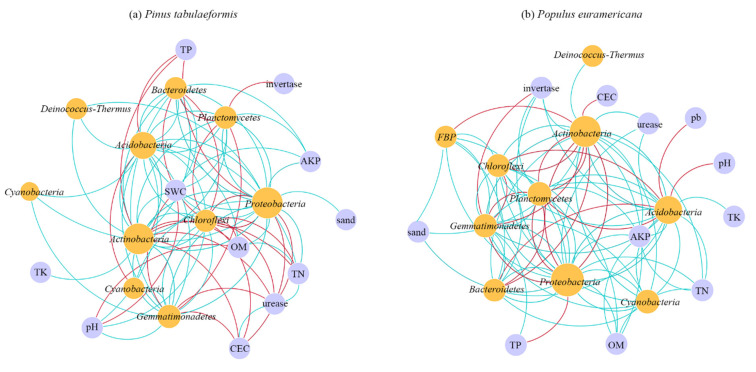
Molecular ecological network of bacterial species composition and environmental factors under *P. tabulaeformis* and *P. euramericana.* Note: AKP, soil alkaline phosphatase activity; CEC, soil cation exchange capacity; OM, soil organic matter; pH, soil pH; TK, soil total potassium; TN, soil total nitrogen; TP, soil total phosphorus; and SWC, soil water content. The blue line represents positive interaction between nodes, and the red line represents negative interaction between nodes.

**Table 1 plants-12-00974-t001:** Information on sampling and control areas.

Site Name	Species	Sample Number	Reclamation Year (a)	Site Name	Species	Sample Number	Reclamation Year (a)
North waste dump	*Pinus tabulaeformis*	N1	15	Inner waste dump	*Pinus tabulaeformis*	I1	9
	*Populus euramericana*	N2	15		*Populus euramericana*	I2	9
East waste dump	*Pinus tabulaeformis*	E1	12	Yinwang waste dump	*Pinus tabulaeformis*	Y1	6
	*Populus euramericana*	E2	12		*Populus euramericana*	Y2	6
West waste dump	*Pinus tabulaeformis*	W1	12	Control area	*Pinus tabulaeformis*	C1	
	*Populus euramericana*	W2	12		*Populus euramericana*	C2	

**Table 2 plants-12-00974-t002:** Comparison of the topological properties of soil bacterial molecular ecological networks under the reclamation of *P. tabulaeformis* and *P. euramericana*.

Network Index	*P. tabulaeformis* (0.960)	*P. euramericana* (0.960)
Total number of nodes	642	746
Total number of connections	1518	2003
Power law R^2^ value	0.701	0.712
Average connectivity	4.729	5.370
Average clustering coefficient	0.541	0.579
Average path distance	9.425	11.582

## Data Availability

Not applicable.

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
