# Peer review of "Comparison of Soil Bacterial Communities under Canopies of *Pinus tabulaeformis* and *Populus euramericana* in a Reclaimed Waste Dump"

_plants, 2023, doi:10.3390/plants12040974_

Round 1
Reviewer 1 Report
Enjoyed reading the manuscript however there are many things that should be changed - to pay attention - Introduction - some more information - do you have a hypothesis ? the names of the plants need to be Italic; need more information about the waste dumps -organic ? or ?; Methods sampling - line 17 plant rhizosphere ? we are talking about a tree - canopy ? had you removed the litter ? etc not clear; results - all the figure need to undergo changes- no title for X and Y axis no marks which one is a, b. etc; Fig 3 should be represented by box plots with SD, no line should connect between the points; line212 no units for the values ?;Fig. 4 titles in chinese etc; All the figures components need to be presented in the figure legend ; the tables are a mix between two languages; line 363 - who is Li Pengfei- Ref ? line 366 Wang Jun ??
In general the goal is nice however - the whole manuscript need to overgo a great revision
Author Response
We are very grateful to your comments for the manuscript. According with your advice, we tried our best to amend the relevant part and made some changes in the manuscript. These changes will not influence the content and framework of the paper. All of your questions were answered below. Here we list the changes.
We appreciate for Reviewers’ warm work earnestly, and hope that the correction will meet with approval.
Point 1: do you have a hypothesis?
Response 1: Thank you very much for your question, our article has a hypothesis that this research is based on a field-scale experiment. The methodology during the land rehabilitation process is consistent, with the sole differentiation being in the techniques used for planting vegetation. The soil cover is sourced from topsoil stripped from open-cut mining, with a thickness of approximately 1.0 meters. The soil is mainly loess, characterized by a relatively infertile, paucity of humus, and has not undergone any processes of maturation.
Point 2: the names of the plants need to be Italic.
Response 2: Thank you for your detailed suggestions. We have made changes based on your comments to italicize the plants.
Point 3: need more information about the waste dumps -organic ? or ?;.
Response 3: Thank you very much for your suggestion. We added the following in response to comments, which you can see in lines 116-119 of the revised version:
The soil cover is sourced from topsoil stripped from open-cut mining, with a thickness of approximately 1.0 m. The soil is mainly loess, characterized by a relatively infertile, paucity of humus, and has not undergone any processes of maturation.
Point 4: Methods sampling - line 17 plant rhizosphere ? we are talking about a tree - canopy ? had you removed the litter ? etc not clear;
Response 4: Your suggestions are much appreciated and we have revised the article title to : Comparison of soil bacterial communities under canopies of Pinus tabulaeformis and Populus euramericana in a reclaimed waste dump.
And we are talking about a tree canopy.
Point 5: results - all the figure need to undergo changes- no title for X and Y axis no marks which one is a, b. etc;
Response 5: Thank you very much for your suggestion. Based on the comments of the specialists, we have changed all figure names.
Point 6: Fig 3 should be represented by box plots with SD, no line should connect between the points;
Response 6: Thank you very much for your suggestion. Based on the comments of the specialists, we have changed figure 3 .
Point 7: line212 no units for the values?
Response 7: Thank you very much for your suggestion, and this section does have some errors. Based on the comments, we have revised the original text to read as follows, see line 232 to 241 of the revised version:
With the increase in reclamation years, the soil bulk density gradually decreased from 2.45 g/cm2 to 2.17 g/cm2, and the proportion of soil sand particles gradually increased from 64.32 % to 75.29 %, while the soil water content had no obvious change. The soil pH gradually decreased from 8.29 to 8.2 with the increase in reclamation years. The cation exchange capacity, soil organic matter, total nitrogen, and total potassium content increased significantly with the increase in reclamation years, from 11.5 coml/kg, 7.09 g/kg, 0.03 %, and 12 g/kg to 14.45 coml/kg, 11.73 g/kg, 0.05 %, and 12.35 g/kg, respectively. The total phosphorus of the soil did not change significantly with the increase in reclamation years. The activities of urease, sucrase, and alkaline phosphatase increased significantly with the increase in reclamation years, from 0.13 mg/g, 8.44 mg/g, and 0.52 mg/g to 0.47 mg/g, 45.26 mg/g, and 0.81 mg/g, respectively.
Point 8: Fig. 4 titles in chinese etc;
Response 8: Thank you very much for your suggestion. Based on the comments of the specialists, we have changed figure 4.
Point 9: All the figures components need to be presented in the figure legend ;
Response 9: Thank you very much for your suggestion. Based on the comments of the specialists, we have changed all figures.
Point 10: the tables are a mix between two languages;
Response 10: Thank you very much for pointing out the problems, in the revised version we have carefully corrected the relevant issues.
Point 11: line 363 - who is Li Pengfei- Ref ? line 366 Wang Jun ?
Response 11: Li Pengfei corresponds to Ref. 16, Wang Jin corresponds to Ref. 17. In the revised version, we noticed this problem and corrected it.
Point 12: In general the goal is nice however - the whole manuscript need to overgo a great revision.
Response 12: Thank you very much for your recognition of our preliminary work and your detailed suggestions on the article's shortcomings. We have responded to your comments and those of other reviewers one by one and carefully revised the issues corresponding to each suggestion, and made a general improvement to the article. We hope that the revised version will be well received by you.

Reviewer 2 Report
Review Hou et al., 2023
In this interesting and original paper by Hou et al, the authors aimed to see if there are differences in the soil bacterial communities of Pinus abulaeformis and Populus euramericana in a reclaimed waste dump. The authors did a good job of results analysis and the correlation between microbes and environmental factors.
In my opinion, the main weakness of the paper is that there is no in-depth discussion in the results of what the differences between the tree species derive from, to what extent there is an effect of the planting time on the soil bacterial communities and what the decrease or increase of certain bacterial phyla means according to various factors of the environment. How do the differences in tree characteristics affect the microbiota? I look forward to an in-depth discussion and reference to the literature on the subject.
Line 414 Where is the statistical analysis for the correlation between bacteria and environmental factors?
Minor:
1. Tree species at Italic.
2. Change the species name from the species to the tree name over the paper it is more useful for the comparison.
3. Line 197, what does the mean “some chemicals” please add the differences.
4. Line 203- Figure 3- Describe the graph in legend. Change the legend language to English. The graph presents a comparison between sites, so Line is not the way to present a Change to a bar graph or other options. Another option, change X axis to reclamation years.
Add the letter of the graph on the graph, for example- (b) soil water content. Add Y axis title. Change all Y axis to 4 data points to follow your described data in the text.
5. To help the readers write the reclamation years in table 1 and figure 3.
6. Line 226 I missed the description of the taxonomic profiling that was identified.
7. Line 233 “P. tabulaeformis is 0.04 higher than that in the reclaimed soil of P. euramericana” What is 0.04? %?
8. Line 236-Figure 4- Change the legend language to English. Bacteria strains at Italic. Relative abundance of bacterial ….-add (%). To help the readers add a name tree to X axis.
9. Line 245-Figure 5 To help the readers add Y axis and letters (a,b,..) to the graphs.
10. Table 2 Change the title language to English
11. Line 441- Figure 10 Please improve the figure and explain it well; for example, change WC to SWC for Soil water content (Also in Figure 9). What is the line colors mean?
Author Response
Response to Reviewer 2 Comments
We are very grateful to your comments for the manuscript. According with your advice, we tried our best to amend the relevant part and made some changes in the manuscript. These changes will not influence the content and framework of the paper. All of your questions were answered below. Here we list the changes.
We appreciate for Reviewers’ warm work earnestly, and hope that the correction will meet with approval.
In my opinion, the main weakness of the paper is that there is no in-depth discussion in the results of what the differences between the tree species derive from, to what extent there is an effect of the planting time on the soil bacterial communities and what the decrease or increase of certain bacterial phyla means according to various factors of the environment. How do the differences in tree characteristics affect the microbiota? I look forward to an in-depth discussion and reference to the literature on the subject.
Response 1: Thank you for your detailed suggestions. We have made changes based on your comments. We have added the relevant text in line 77-103, 232-239, 270-285, 537-540 of the revised version. However, the differences between the tree species derive from has not been studied in depth, and the mechanism will continue to be studied in the future.
Line 414 Where is the statistical analysis for the correlation between bacteria and environmental factors?
Response 2: Thank you for your detailed suggestions. The statistical analysis for the correlation between bacteria and environmental factors would be studied in the future.
Minor:
Point 1: Tree species at Italic
Response 1: Thank you for your detailed suggestions. We have made changes based on your comments to italicize the plants.
Point 2: Change the species name from the species to the tree name over the paper it is more useful for the comparison.
Response 2: Thank you for your detailed suggestions. We have made changes based on your comments.
Point 3: Line 197, what does the mean “some chemicals” please add the differences.
Response 3: Thank you very much for your suggestion. We have added the relevant text in line 203 of the revised version:
The soil physical and chemical properties of different reclamation tree species are shown in Figure 3. There are certain differences in soil physical and chemical properties and enzyme ac-tivities under P. tabulaeformis and P. euramericana. There is no significant difference in the physical properties of the soil under the two types of plants. The average soil water content un-der P. tabulaeformis is 1.12 % higher than that under P. Euramericana, while the average values of bulk density and sand content under P. tabulaeformis are 0.05 g/cm2 and 2.63 % lower than that under P. euramericana. There is no obvious difference in some chemical properties of the soil under the two types of plants. The average pH value and average cation exchange capacity (CEC) of the soil under P. tabulaeformis are higher than those of P. euramericana by 0.01 and 1.25 coml/kg, respectively, while the average total nitrogen, total phosphorus, total potassium, and soil organic matter contents are relatively low by 0.01 %, 0.06 g/kg, 0.33 g/kg, and 1.89 g/kg, respectively. The activities of soil urease, invertasesucrase, and alkaline phosphatase under P. tabulaeformis are 0.18 mg/g, 3.72 mg/g, and 0.22 mg/g lower than those under P. euramericana, respectively. After 15 years of reclamation, the soil organic matter content and urease activity were significantly different from the control group, and bulk density, soil water content, pH, cation exchange capacity, total nitrogen, total phosphorus, total potassium, and invertase activ-ity were basically close to the control group. In short, with the increase of reclamation years, the reclaimed soil properties were close to the control group, and the soil properties of Pinus tabu-laeformis were closer to the control group than that of Populus euramericana.
Point 4: Line 203- Figure 3- Describe the graph in legend. Change the legend language to English. The graph presents a comparison between sites, so Line is not the way to present a Change to a bar graph or other options. Another option, change X axis to reclamation years.
Response 4: Thank you very much for your suggestion. According to the suggestion, we have added the introduction of reclamation years in table 1, so we will not repeat the description in figure 3.
Point 5: Add the letter of the graph on the graph, for example- (b) soil water content. Add Y axis title. Change all Y axis to 4 data points to follow your described data in the text.
Response 5: Thank you very much for your suggestion. We have made changes based on the comments, see fig. 3 of the revised version.
Point 6: Line 226 I missed the description of the taxonomic profiling that was identified.
Response 6: Thank you very much for your suggestion. We have made changes based on the comments, see line 252 to 265 of the revised version:
The relative abundance of Proteobacteria was the largest, ranging from 27.34 % to 39.34 %. On in 12 a and 15 a of reclamation, the relative abundance of Proteobacteria in P. euramericana was higher than that of P. tabulaeformis. However, the relative abundance of P. tabulaeformis was lower than that of P. euramericana in 6 a and 9 a of reclamation. Proteo-bacteria participated in the soil nitrogen cycle process and help to restore the improvement of soil quality. The relative abundance of Actinobacteria was 16.55 % - 36.82 %, which was in-creasing with the increase of reclamation years. Actinobacteria was a saprophytic aerobic bac-terium that preferred neutral or slightly alkaline, promoting the growth of rhizosphere bacteria, symbiotic bacteria and endophytic bacteria. The relative abundance of Acidobacterium was 7.32 % - 17.12 %. In the samples of P. tabulaeformis, the abundance was increasing with the in-crease of reclamation years. This bacterium played an important role in the degradation of plant residues. The abundance of Chloroflexi in P. tabulaeformis samples decreased with the increase of reclamation years. The nutrition mode of Chloroflexi was extremely diverse.
Point 7: Line 233 “P. tabulaeformis is 0.04 higher than that in the reclaimed soil of P. euramericana” What is 0.04? %?
Response 7: Thank you very much for your suggestion. We have made changes based on the comments, see line 266 to 270 of the revised version:
From the perspective of vegetation type, at the phylum level, the relative abundance of Actinobacteria in the reclaimed soil of P. tabulaeformis is 0.04 % higher than that in the reclaimed soil of P. euramericana. Furthermore, the relative abundance of Acidobacteria and Cyanobacteria in the reclaimed soil of P. tabulaeformis is 0.02 % lower than that in the reclaimed soil of P. euramericana.
Point 8: Line 236-Figure 4- Change the legend language to English. Bacteria strains at Italic.
Response 8: Thank you very much for your suggestion. We have made changes based on the comments, see fig. 4 of the revised version.
Point 9: Line 245-Figure 5 To help the readers add Y axis and letters (a,b,..) to the graphs.
Response 9: Thank you very much for your suggestion. We have made changes based on the comments, see fig. 5 of the revised version.
Point 10: Table 2 Change the title language to English.
Response 10: Thank you very much for your suggestion. We have made changes based on the comments, see Table 2 of the revised version.
Point 11: Line 441- Figure 10 Please improve the figure and explain it well; for example, change WC to SWC for Soil water content (Also in Figure 9). What is the line colors mean?
Response 11: Thank you very much for your suggestion. We have made changes based on the comments, see fig. 9 and 10 of the revised version.
At the same time, WC modefies SWC. The blue line represents positive interaction between nodes, and the red line represents negative interaction between nodes.

Reviewer 3 Report
This paper deals with an interesting problem concerning the influence of two different (deciduous, coniferous) species on the formation of bacterial communities on a reclaimed anthropogenic site. The work is essential and relevant from a practical point of view. It is written in a very disorganised manner and needs to be reorganised. Some of the below:
1. Include the name of the species authors (Pinus tabulaeformis Carr.; Populus euramericana (Dode) Guinier) in the title, and it is to be in italics.
2. May it be a better title? Comparison of soil bacterial communities under canopies of Pinus tabulaeformis Carr. and Populus euramericana (Dode) Guinier) in a reclaimed waste dump
3. The abstract is written chaotically and needs to be reorganised. All Latin names of plants and families of bacteria need to be written in italics (throughout the text).
4. The Introduction needs to be improved and supplemented by other work worldwide. Such information is currently lacking. At the end of the Introduction, there should be a clear statement of the aims of the work; now, it is not clear where this is.
5. The location map is poor quality, and the geographical coordinates were incorrectly written.
Please, give the international names of chestnut soil.
The location of the sampling site is badly present.
In Table 1, change vegetation to species.
automatic Kjeldahl nitrogen meter??????
Figure 1 is illegible and unintelligible, even the Chinese letters and the low quality of the drawing—the same situation with Figure 5.
Line 22-225 - same title section???
Table 2 is partly with Chines letters, so it isn't easy to understand.
Conclusions: Despite the interesting subject matter and credible results, their presentation has much to be desired. The work needs to be reorganised: the study area needs to be revised, and the research sites need to be stated and compared in tables. Titles of figures do not always correspond to their content. While the topic is appropriate and fits within the journal's scope, the authors ignore the editorial requirement of the journal (although this is not important at the review level)
Author Response
We are very grateful to your comments for the manuscript. According with your advice, we tried our best to amend the relevant part and made some changes in the manuscript. These changes will not influence the content and framework of the paper. All of your questions were answered below. Here we list the changes.
We appreciate for Reviewers’ warm work earnestly, and hope that the correction will meet with approval.
Point 1: Include the name of the species authors (Pinus tabulaeformis Carr.; Populus euramericana (Dode) Guinier) in the title, and it is to be in italics.
Response 1: Thank you very much for your suggestion, we have re-written the name in italics.
Point 2: May it be a better title? Comparison of soil bacterial communities under canopies of Pinus tabulaeformis Carr. and Populus euramericana (Dode) Guinier) in a reclaimed waste dump.
Response 2: We agree with your suggestion. We have revised the article title to : Comparison of soil bacterial communities under canopies of Pinus tabulaeformis and Populus euramericana in a reclaimed waste dump.
Point 3: The abstract is written chaotically and needs to be reorganised. All Latin names of plants and families of bacteria need to be written in italics (throughout the text).
Response 3: Thank you very much for your suggestion. We have reorganized the summary section based on your suggestions, as shown in line 19 of the revised version:
Abstract: To compare the effects of different remediation tree species on soil bacterial communities and provide a theoretical basis for the selection of ecosystem function promotion strategies after vegetation restoration, the characteristic changes in soil bacterial communities after Pinus tabulaeformis and Populus euramericana reclamation were explored using high-throughput sequencing and molecular ecological network methods. The results showed that 1) With the increase of reclamation years, the reclaimed soil properties were close to the control group, and the soil properties of Pinus tabulaeformis were closer to the control group than that of P. euramericana. 1) The dominant bacteria under canopies of P. tabulaeformis and Populus euramericana was the same. Proteobacteria, Actinobacteria, Acidobacteria, Chloroflexi, Gemmatimonadetes, Planctomycetes, Bacteroidetes, and Cyanobacteria were the dominant bacteria in the restored soil, accounting for more than 95% of the total abundance. The average values of the Shannon diversity index, Simpson diversity index, Chao 1 richness estimator, and abundance-based coverage estimator of the bacterial community in the P. euramericana reclaimed soil were higher than those in the P. tabulaeformis reclaimed soil.The influence of reclamation years on the bacterial community of samples is greater than that of species types. 3) The results of ecological network construction showed that the total number of nodes, total number of connections and average connectivity of soil bacterial network under P. euramericana reclamation were greater than those under P. tabulaeformis reclamation. The bacterial molecular ecological network under P. euramericana was more abundant. 4) Among the dominant bacteria, the relative abundance of Actinobacteria was negatively correlated with soil pH, soil total nitrogen content, and the activities of urease, invertase, and alkaline phosphatase, while the relative abundance of Proteobacteria and Bacteroidetes was positively correlated with these environmental factors. The relationship between the soil bacterial community of P. tabu-laeformis and P. euramericana and the environmental factors is not completely the same, and even the interaction between some environmental factors and bacteria is opposite.
Point 4: The Introduction needs to be improved and supplemented by other work worldwide. Such information is currently lacking. At the end of the Introduction, there should be a clear statement of the aims of the work; now, it is not clear where this is.
Response 4: Thank you very much for your suggestion. We have made changes based on the comments, see introduction of the revised version.
Point 5: The location map is poor quality, and the geographical coordinates were incorrectly written.
Response 5: Thank you very much for your suggestion.We have modified the issues related to the location map, see fig. 1 of the revised version.
Point 6: Please, give the international names of chestnut soil.
Response 6: We are very sorry for not using international names. After checking, the international name of chestnut soil should be castanozems.
Point 7: The location of the sampling site is badly present.
Response 7: Thank you very much for your suggestion. We have made changes based on the comments, see Figure 1 of the revised version.
Point 8: In Table 1, change vegetation to species.
Response 8: Thank you very much for your suggestion. We have made changes based on the comments, see Table 1 of the revised version.
Point 9: automatic Kjeldahl nitrogen meter??????
Response 9: Thank you very much for pointing out the problems, the correct way to write it should be: Automatic Kjeldahl Nitrogen Meter.
Point 10: Figure 1 is illegible and unintelligible, even the Chinese letters and the low quality of the drawing—the same situation with Figure 5.
Response 10: Thank you very much for pointing out the problems, we redrew Figure 1 and Figure 5, modifying the related issues.
Point 11: Line 22-225 - same title section???
Response 11: Thank you very much for pointing out the problems, we corrected this issue in the revised version.
Point 12: Table 2 is partly with Chines letters, so it isn't easy to understand.
Response 12: Thank you very much for your suggestion. We have made changes based on the comments, see Table 2 of the revised version.

Round 2
Reviewer 3 Report
Good job